**Citation:** *Molecular Systems Biology* 9:700
www.molecularsystemsbiology.com

# Bacterial evolution of antibiotic hypersensitivity

Viktória Lázár[1], Gajinder Pal Singh[1], Réka Spohn[1], István Nagy[2], Balázs Horváth[2], Mónika Hrtyan[1], Róbert Busa-Fekete[3], Balázs Bogos[1], Orsolya Méhi[1], Bálint Csörgő[1], György Pósfai[1], Gergely Fekete[1], Balázs Szappanos[1], Balázs Kégl[3], Balázs Papp[1,*] and Csaba Pál[1,*]

[1] Synthetic and Systems Biology Unit, Institute of Biochemistry, Biological Research Center, Szeged, Hungary, [2] Genomics Unit, Institute of Biochemistry, Biological Research Center, Szeged, Hungary and [3] Linear Accelerator Laboratory, University of Paris-Sud, CNRS, Orsay, France
* Corresponding authors. C Pál or B Papp, Synthetic and Systems Biology Unit, Institute of Biochemistry, Biological Research Center, Temesvari krt 62, Szeged 6726, Hungary. Tel.: +36 62 599 661; Fax: +36 62 433 506; E-mail: cpal@brc.hu or pappb@brc.hu

The evolution of resistance to a single antibiotic is frequently accompanied by increased resistance to multiple other antimicrobial agents. In sharp contrast, very little is known about the frequency and mechanisms underlying collateral sensitivity. In this case, genetic adaptation under antibiotic stress yields enhanced sensitivity to other antibiotics. Using large-scale laboratory evolutionary experiments with *Escherichia coli*, we demonstrate that collateral sensitivity occurs frequently during the evolution of antibiotic resistance. Specifically, populations adapted to aminoglycosides have an especially low fitness in the presence of several other antibiotics. Whole-genome sequencing of laboratory-evolved strains revealed multiple mechanisms underlying aminoglycoside resistance, including a reduction in the proton-motive force (PMF) across the inner membrane. We propose that as a side effect, these mutations diminish the activity of PMF-dependent major efflux pumps (including the *AcrAB* transporter), leading to hypersensitivity to several other antibiotics. More generally, our work offers an insight into the mechanisms that drive the evolution of negative trade-offs under antibiotic selection.
*Molecular Systems Biology* **9**: 700; published online 29 October 2013; doi:10.1038/msb.2013.57
*Subject Categories:* microbiology & pathogens
*Keywords:* antibiotic resistance; collateral sensitivity network; evolutionary experiment; trade off

## Introduction

Evolutionary adaptation to an environment may be accompanied by a decline or an increase in fitness in other environments. Although such trade-offs are frequently observed in nature, the governing rules and the corresponding molecular mechanisms are generally unclear. The evolution of antibiotic resistance offers an ideal model system to systematically investigate this issue. Enhanced level of resistance can be achieved by mutations in the genome or by acquisition of resistance-conferring genes through horizontal gene transfer. The relative contribution of these mechanisms depends both on the antibiotic employed and on the bacterial species considered (Alekshun and Levy, 2007). It has been suggested that the progressive accumulation of mutations can simultaneously change an organism's sensitivity to many different antimicrobial agents and can serve as the first step in the evolution of clinically significant resistance by more specific and effective mechanisms (Baquero, 2001; Goldstein, 2007; Gullberg *et al*, 2011). A recent review argued that the evolution of multidrug resistance and hypersensitivity are among the central issues of the field (Palmer and Kishony, 2013). Better understanding of these phenomena is important as they could potentially inform future therapeutic strategies to mitigate resistance evolution. For example, the choice of

optimal antibiotic combinations depends on both the presence of physiological drug interactions and the frequency of mutations with pleiotropic fitness effects (Chait *et al*, 2007; Palmer and Kishony, 2013; Pena-Miller *et al*, 2013).

Specifically, it remains unclear how frequently genetic adaptation to a single antibiotic increases the sensitivity to others and what the underlying molecular mechanisms of hypersensitivity are. No large-scale, systematic laboratory evolution study has been devoted to investigate this problem under controlled environmental settings. To our best knowledge, the only prior work with similar aims was published 60 years ago and was limited to phenomenological descriptions (Szybalski and Bryson, 1952).

Here, for the first time, we apply an integrated approach to decipher collateral-sensitivity interactions between antibiotics. We initiated the laboratory evolution of *E. coli* populations in the presence of one of the several different antimicrobial agents. These antibiotics are well characterized, widely employed in the clinic, and have diverse modes of actions (Table I). Our list also includes antibiotics that are typically used against Gram-positive bacteria. Consistent with previous studies (Curtiss *et al*, 1965; Vuorio and Vaara, 1992; Elkins and Nikaido, 2002), we found that these antibiotics inhibited the growth of wild-type *E. coli* at high concentrations and that resistance readily evolved against these compounds

**Table I** Employed antibiotics and their modes of actions

| Antibiotic name | Abbreviation | Mode of action | Bactericidal or Bacteriostatic |
|---|---|---|---|
| Ampicillin | AMP* | Cell wall | Bactericidal |
| Pipericallin | PIP | Cell wall | Bactericidal |
| Cefoxitin | FOX* | Cell wall | Bactericidal |
| Fosfomycin | FOS | Cell wall | Bactericidal |
| Lomefloxacin | LOM | Gyrase | Bactericidal |
| Ciprofloxacin | CPR* | Gyrase | Bactericidal |
| Nalidixic acid | NAL* | Gyrase | Bactericidal |
| Fosmidomycin | FSM | Lipid | Bactericidal |
| Nitrofurantoin | NIT* | Multiple mechanisms | Bactericidal |
| Amikacin | AMK | Aminoglycoside | Bactericidal |
| Gentamicin | GEN | Aminoglycoside | Bactericidal |
| Kanamycin | KAN* | Aminoglycoside | Bactericidal |
| Tobramycin | TOB* | Aminoglycoside | Bactericidal |
| Streptomycin | STR | Aminoglycoside | Bactericidal |
| Tetracycline | TET* | Protein synthesis, 30S | Bacteriostatic |
| Doxycycline | DOX* | Protein synthesis, 30S | Bacteriostatic |
| Chloramphenicol | CHL* | Protein synthesis, 50S | Bacteriostatic |
| Erythromycin | ERY* | Protein synthesis, 50S | Bacteriostatic |
| Fusidic acid | FUS | Protein synthesis, 50S | Bacteriostatic |
| Sulfamonomethoxine | SLF | Folic acid biosynthesis | Bacteriostatic |
| Trimethoprim | TRM* | Folic acid biosynthesis | Bacteriostatic |
| Muporicin | MUP | Gram positive | NA |
| Cycloserine | CYC | Gram positive | NA |
| Vancomycin | VAN | Gram positive | NA |

The functional classification is based on previous studies (Girgis *et al*, 2009; Yeh *et al*, 2006). For an overlapping set of 12 selected antibiotics (indicated by stars), populations were allowed to evolve in the presence of successively increased antibiotic concentrations.

(see below). Next, we charted the network of collateral-sensitivity interactions by measuring the susceptibility of each evolved line against all the other antibiotics. Our analysis revealed a strikingly dense network of collateral-sensitivity interactions. Many of these interactions involved aminoglycosides. Finally, laboratory-evolved lines were subjected to whole-genome sequence analysis and biochemical assays to decipher the underlying molecular mechanisms of these interactions.

## Results

### Parallel evolution of antibiotic susceptibility patterns in the laboratory

We followed established protocols with minor modifications to evolve bacterial populations under controlled laboratory conditions (Hegreness *et al*, 2008). Starting from a single ancestral clone, populations were propagated in batch culture (minimal glucose medium containing a single antibiotic), whereby 1% of each culture was diluted into fresh medium on a daily basis.

Microbes frequently encounter low or varying antibiotic concentrations (Baquero, 2001). For example, the limited accessibility of antibiotics to certain tissues or incomplete treatment can lead to the formation of concentration gradients within the body (Kohanski *et al*, 2010a). Similarly, antibiotic-polluted natural environments generally form such gradients radiating from the source. To mimic these natural conditions, we employed two selection regimes. In the first set of experiments, a fixed sublethal antibiotic concentration (i.e., sufficient to reduce the growth of the starting population by 50%) was employed. Using this set-up, we propagated 10 independent populations in the presence of each antibiotic for ∼140 generations, resulting in 240 parallel-evolved lines. As selection pressure and resistance-conferring mutations can differ substantially between low and high antibiotic concentrations, we also employed a complementary laboratory evolutionary protocol. For an overlapping set of 12 selected antibiotics (Table I), populations were allowed to evolve to successively higher antibiotic concentrations (96 replicate populations per antibiotic). Starting with subinhibitory antibiotic concentrations, the antibiotic concentration was increased every 4 days over a period of 240–384 generations. Despite the short evolutionary timescale, many of the evolved populations reached very high resistance levels (20- to 328-fold increases in the minimum inhibitory concentrations (MICs); Supplementary Table S1). For each antibiotic, we selected 10 independently evolved resistant populations for further analysis (Materials and methods). In addition, to control for potential adaptive changes that are not specific to the employed antibiotics, we also established 10 parallel populations that were grown in an environment devoid of antibiotics, referred to as adapted control lines.

Next, we measured the corresponding changes in the sensitivities of all laboratory-evolved populations to other antibiotics. Fitness was measured by monitoring the optical density of liquid cultures of all evolved and control lines in the presence and absence of sublethal concentrations of antibiotics. Our protocol was highly sensitive and could efficiently detect both weak negative and positive trade-offs, which may be overlooked in other assays (Materials and methods; Supplementary Text S1). Furthermore, by measuring fitness in antibiotic-free medium, we could distinguish between general growth defects and genuine collateral-sensitivity interactions. Specifically, we employed a rigorous statistical procedure to identify those collateral-sensitivity interactions that are not expected based on the generally weak growth defect observed in the absence of antibiotics (see Supplementary Text S2 and Supplementary Figure S3). The reliability of the method was confirmed by comparing its results with sensitivity estimates based on the colony size (Supplementary Text S2; Supplementary Figure S1).

We noticed that parallel-evolving populations exposed to the same antibiotic displayed very similar antibiotic susceptibility patterns (Supplementary Figure S2). Thus, we developed a data analysis pipeline to infer evolutionary interactions at the level of antibiotic pairs based on the growth patterns of the antibiotic-adapted and control populations. The analysis ultimately led to a map of evolutionary interactions between antibiotics (Figures 1A and B; Supplementary Table S2). In this study, we concentrated on antibiotic pairs showing collateral sensitivity; cross-resistance interactions will be described elsewhere.

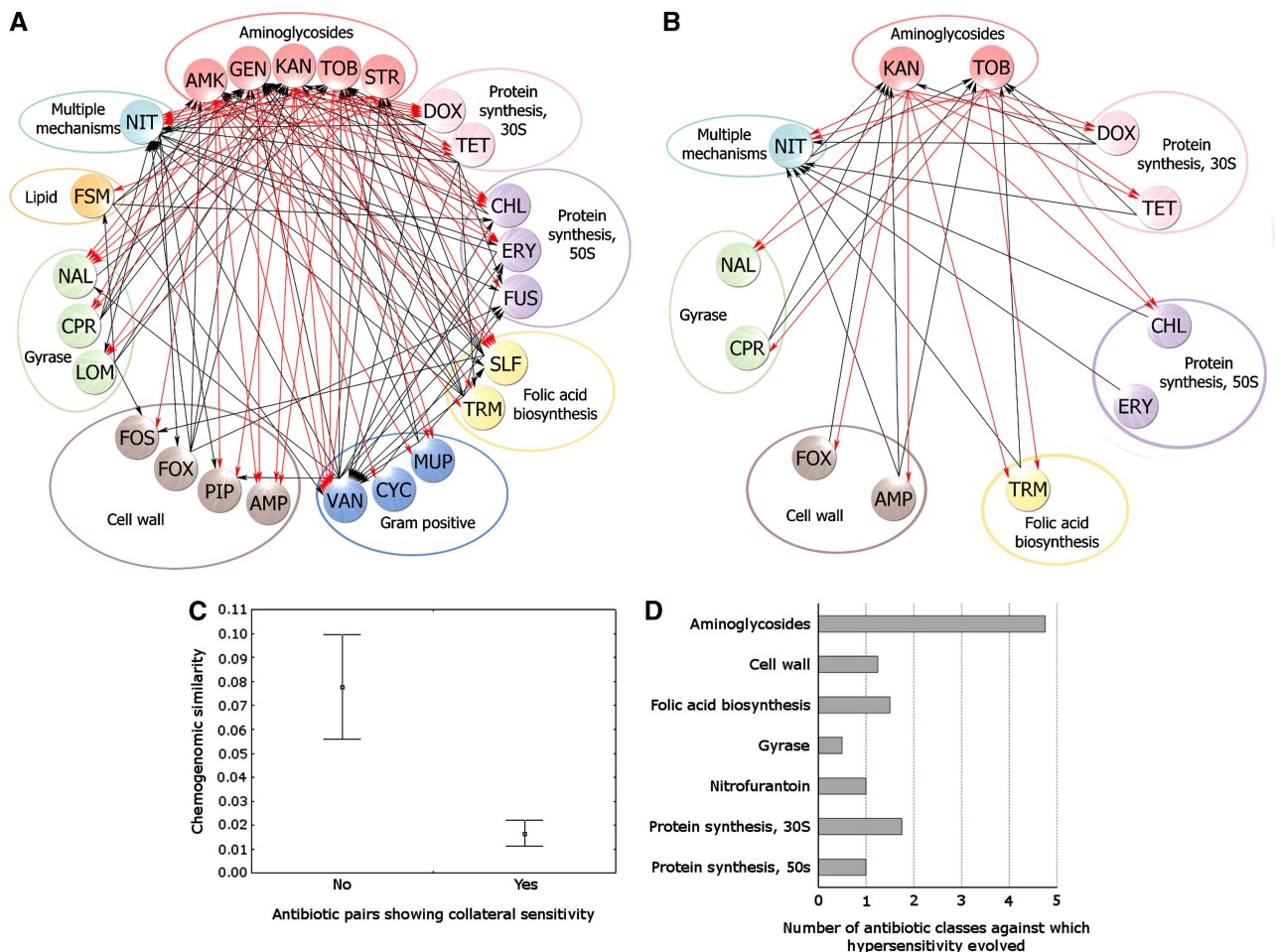

**Figure 1** Networks of collateral-sensitivity interactions. Collateral-sensitivity interaction networks inferred from the adaptation to (**A**) low antibiotic concentrations and (**B**) increasing concentrations of antibiotics. Antibiotics are grouped according to their mode of action. An arrow from antibiotic A to antibiotic B indicates that adaptation to A increased the sensitivity to B. Aminoglycosides dominate the collateral-sensitivity network, with numerous links to other classes of antibiotics (red arrows). (**C**) Collateral-sensitivity antibiotic pairs show relatively low overlap in their chemogenomic profiles ($N = 120$, Mann–Whitney U-test $P < 10^{-5}$). Chemogenomic distance was calculated as pairwise Jaccard distance between sets of genes that influence antibiotic susceptibility (Girgis *et al*, 2009). Error bars indicate 95% confidence intervals. (**D**) Collateral-sensitivity interaction degrees of antibiotic classes (i.e., average number of antibiotic classes against which a population evolves hypersensitivity if exposed to the antibiotic class shown on the vertical axis. Degrees are weighted by the number of antibiotics representing each class).

## Uneven distribution of collateral sensitivity across antibiotic classes

The maps based on the evolutionary experiments performed with constant and gradually increasing antibiotic concentrations were similar. In all, 85% of the interactions between antibiotics overlapped ($P < 10^{-5}$, randomization test). Three main patterns emerge from our map. First, these interactions occurred frequently: at least 35% of all investigated antibiotic pairs showed collateral sensitivity in at least one direction. Second, the mode of antibiotic action has a strong influence on the distribution of interactions. Collateral sensitivity never occurred between antibiotic pairs that target the same cellular subsystem (Fisher's exact test, $P = 0.013$). Thanks to systematic chemogenomic studies, the mode of antibiotic action can be defined and compared in a more quantitative manner. Specifically, a previous study exposed a nearly complete mutagenized *E. coli* library to several antibiotics and determined the fitness contribution of individual genes

(Girgis *et al*, 2009). Using this data set, we calculated the sets of genes that influence susceptibility for each antibiotic used in our study (chemogenomic profile). Collateral sensitivity was depleted between antibiotic pairs with substantial overlap in their chemogenomic profiles (Figure 1C). Third, most antibiotic classes displayed collateral sensitivity with relatively few other classes (Figure 1D). There was one major exception: 44% of the collateral-sensitivity interactions involved aminoglycosides. Genetic adaptation to aminoglycosides increased the sensitivity to many other classes of antibiotics, including inhibitors of DNA synthesis, cell-wall synthesis, and other classes of protein synthesis inhibitors. The observed interactions generally represented 2- to 10-fold decreases in the MICs (Figure 2; Supplementary Table S3), a result that is consistent with an earlier report on antibiotic hypersensitivity (Szybalski and Bryson, 1952). This rate is also rather similar to the 2- to 8-fold increases in MIC typically observed in different efflux pump mutants (Piddock, 2006).

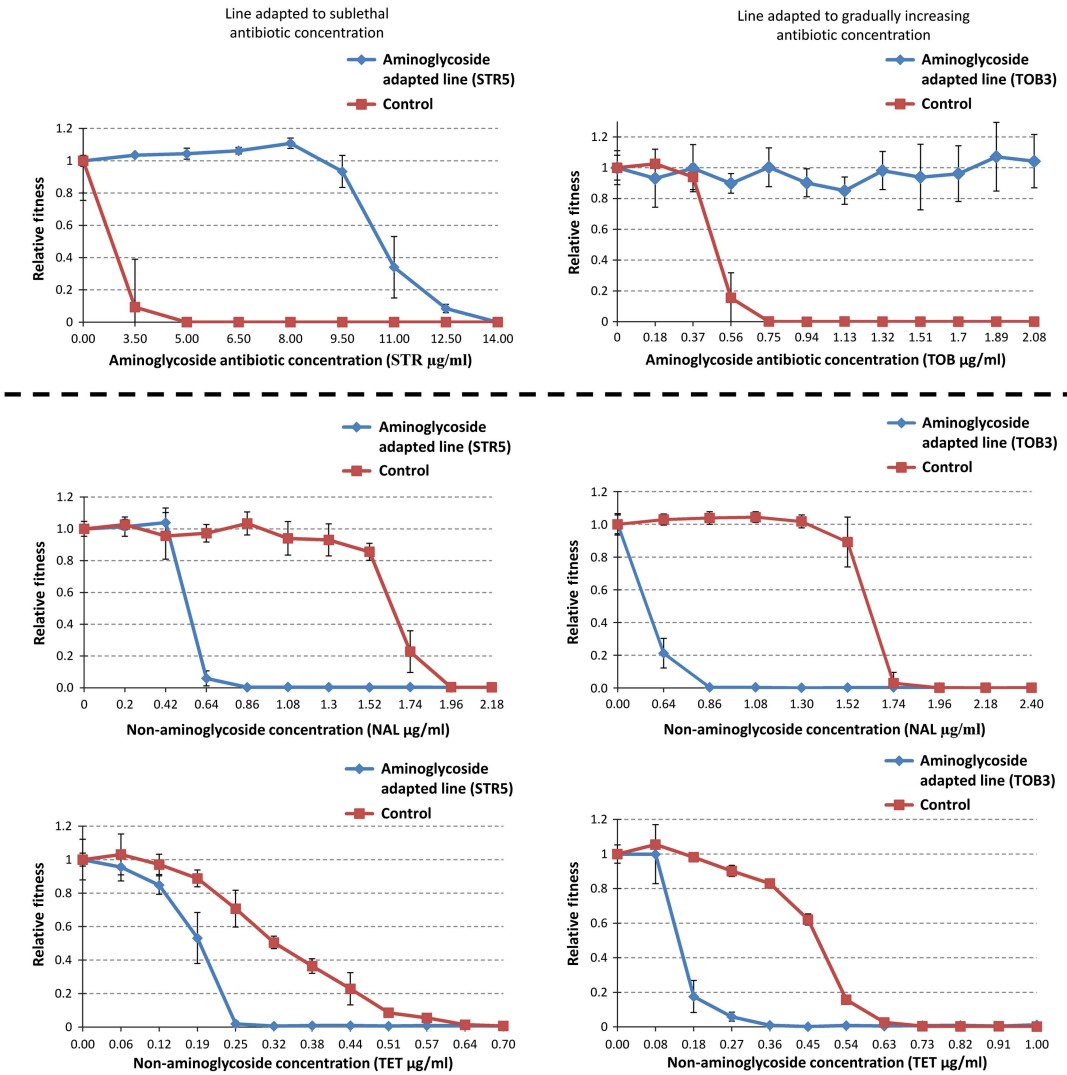

**Figure 2** Dose–response curve of selected aminoglycoside-adapted lineages exhibiting collateral sensitivity. Error bars indicate 95% confidence intervals.

## Multiple mechanisms underlying aminoglycoside resistance

Three major mechanisms of aminoglycoside resistance have been recognized: inactivation of the drugs by aminoglycoside modifying enzymes, modification of ribosome, and decreased membrane permeability (partly through changes in a membrane potential). To gain insight into the molecular mechanisms underlying aminoglycoside resistance and collateral sensitivity in our laboratory-evolved strains, we selected 14 clones evolved in the presence of a single aminoglycoside (kanamycin, tobramycin, or streptomycin) for whole-genome resequencing. All of these clones exhibited hypersensitivity to other classes of antibiotics (Supplementary Tables S2 and S3). The genomes of these independently evolved clones were resequenced using the Applied Biosystems SOLiD platform, and the identified single-nucleotide polymorphisms (SNPs) were confirmed using capillary sequencing.

In total, we identified 100 mutations (SNPs and indels) affecting 44 protein-coding genes. On average, we observed eight mutations per clone in lines adapted to increasing concentrations and two mutations in those adapted to a fixed sublethal concentration (Supplementary Table S4). Three lines of evidence indicated that these substitutions were driven by adaptive evolution. First, 89% of the mutations were in protein-coding regions and were non-synonymous. Second, convergent evolution was prevalent at multiple levels, as a total of 6.7% of the mutations at the single nucleotide level were shared by two or more clones (Supplementary Table S4). Evolutionary convergence was even more apparent at the level of genes and functional units, as a total of 29.5% of the affected 44 genes were mutated repeatedly (Supplementary Tables S4 and S5). Third, comparison with the results of available chemogenomic screens revealed that 36% of the mutated genes influence aminoglycoside susceptibility when inactivated (Supplementary Table S4, $P = 0.013$, Fisher's exact test).

Aminoglycosides directly target the ribosome. Mistranslation and the consequent misfolding of membrane proteins have an important role in aminoglycoside-induced oxidative stress and cell death (Kohanski *et al*, 2010b). Aminoglycosides

generally require respiration for uptake (Taber *et al*, 1987) and enter the cell in a membrane potential-dependent manner. This process relies on cytochromes and the maintenance of the proton-motive force (PMF) through the quinone pool (Kohanski *et al*, 2010b).

Pathway enrichment analyses (Carbon *et al*, 2009) revealed the overrepresentation of several biological processes in the set of accumulated mutations (Supplementary Table S6). In agreement with our expectations, one major target of selection was the translational machinery, including several ribosomal proteins, elongation factors (fusA, rpsL), and tRNA synthetases. Second, several genes involved in membrane transport, phospholipid synthesis, and cell envelope homeostasis were mutated. Remarkably, this list included an oligopeptide transporter (OppF) with a key role in the recycling of cell-wall peptides and the two-component stress-response sensor CpxA (Kohanski *et al*, 2008; Supplementary Table S4). The biosynthesis of polyamines (including putrescine and spermidine) was also affected. These molecules reduce the intracellular production of reactive oxygen species during aminoglycoside stress (Tkachenko *et al*, 2012) and thereby diminish the levels of protein and DNA damage (Kohanski *et al*, 2010b). Third, we identified a broad class of genes expected to influence the membrane electrochemical potential (Supplementary Tables S4 and S5; Figure 2B). These genes are involved in oxidative phosphorylation, proton-potassium symport (*trkH*), oxygen-binding heme biosynthesis (*hemA*), while others are members of the cytochrome terminal oxidase complex (*cyoB*, *cyoC*). They also frequently affect the quinone pool, which serve as electron carriers in the respiratory electron transport chain (IspA and the Nuo protein complex). This third class most likely has a central contribution to the collateral-sensitivity patterns observed, not least because all sequenced clones had at least one mutation in this subsystem (Supplementary Table S4).

## Evidence for antagonistic mutational effects on membrane permeability

Why should membrane potential affecting mutations alter the susceptibility to multiple different antibiotics? These genes are expected to influence aminoglycoside-induced oxidative stress and/or aminoglycoside uptake. Indeed, aminoglycosides uniquely require the PMF for active cellular uptake (Taber *et al*, 1987; Allison *et al*, 2011). In sharp contrast, the efflux of many other antibiotics depends on PMF-dependent pumps (Paulsen *et al*, 1996). On the basis of these observations, we propose a model in which low-level aminoglycoside resistance is achieved by altering the membrane potential across the inner bacterial membrane (Figure 3). As a secondary consequence, these mutations diminish the activity of PMF-dependent major efflux pumps. Indeed, it has been previously shown that CCCP, a chemical inhibitor of oxidative phosphorylation, decreases the intracellular accumulation of aminoglycosides (Allison *et al*, 2011), but most likely increases the intracellular accumulation of several other antibiotics (Coldham *et al*, 2010).

Along with the observed mutations, biochemical assays provided further support for this model. First, we investigated changes in the membrane potential in aminoglycoside-

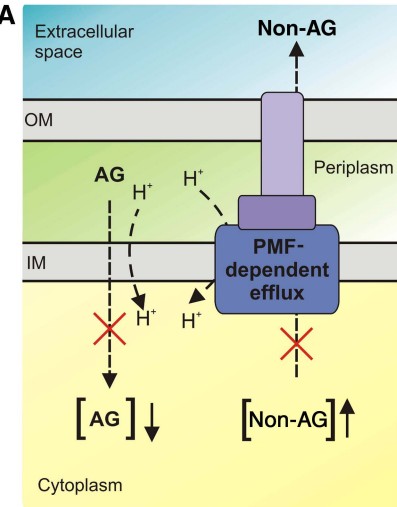

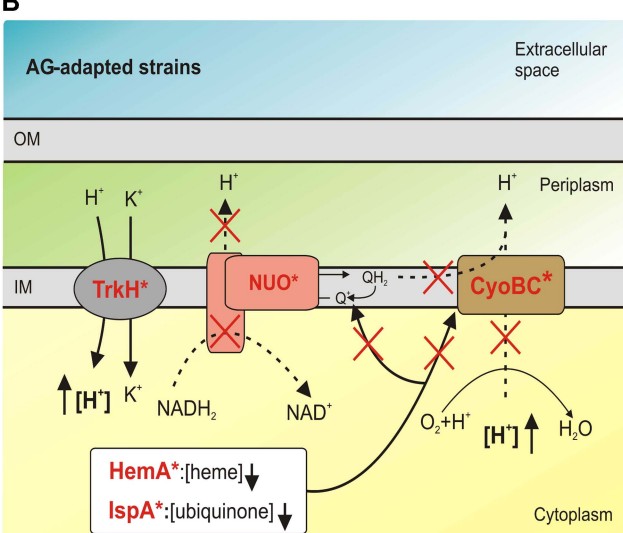

**Figure 3** A putative mechanism underlying collateral sensitivity. (**A**) The theory. Altering the membrane potential across the inner bacterial membrane has two opposing effects: it reduces the uptake of many aminoglycoside-related antibiotics but simultaneously may lead to the reduced activity of PMF-dependent efflux pumps. For more details, see the main text. (**B**) Mutations supporting the theory. Whole-genome sequencing revealed that adaptation to aminoglycosides frequently proceeds through mutations that most likely diminish the generation of the PMF. Mutations are indicated by red, bolded protein names (TrkH, CyoB, HemA, IspA). The observed mutations in TrkH most likely increase the proton influx, whereas the mutations in CyoB and HemA (resulting in the inhibition of proton translocation and heme biosynthesis, respectively) interfere with the proper functioning of the cytochrome terminal oxidase complex. Furthermore, decreased IspA activity reduces the levels of membrane-bound quinones and therefore the level of oxidative phosphorylation. Altogether, these mutations likely reduce the PMF and thus aminoglycoside uptake. Simultaneously, the activity of the PMF-dependent efflux system is expected to decrease, resulting in greater sensitivity to antibiotics transported by these pumps. AG, aminoglycoside; OM, outer membrane; IM, inner membrane; NUO, NADH-Ubiquinone-oxidoreductase; PMF, proton-motive force.

resistant strains. The membrane potential was monitored using the carbocyanine dye diethyloxacarbocyanine (DiOC2) (3) (Novo *et al*, 2000). In agreement with our expectations, the membrane potential was reduced in aminoglycoside-adapted populations (Figure 4A; Supplementary Figure S4;

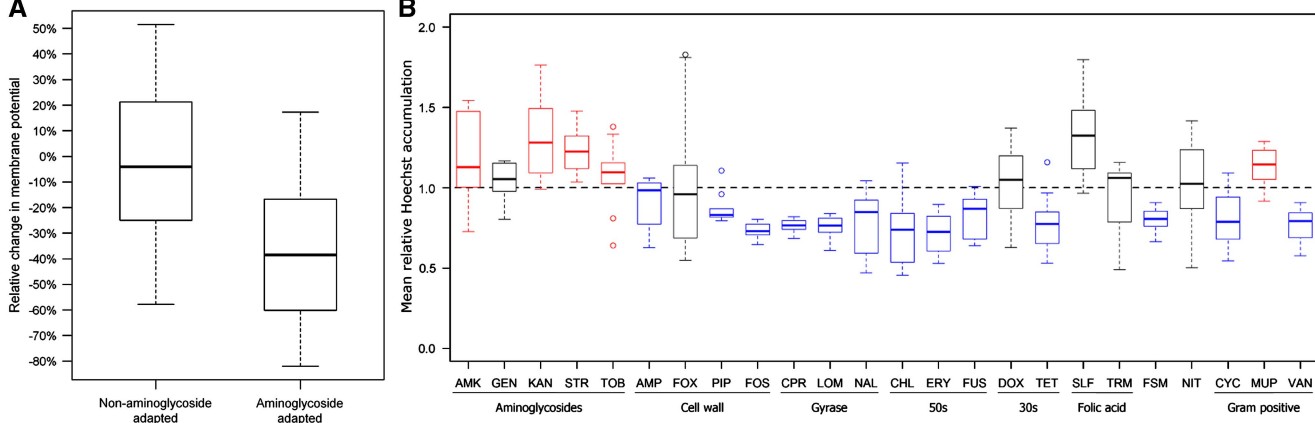

**Figure 4** Membrane permeability (Hoechst dye) and membrane potential changes in evolved lines. (**A**) Membrane potential changes in antibiotic-adapted populations. Changes in the membrane potential were monitored using the carbocyanine dye DiOC2(3). The red/green fluorescence values for a representative set of aminoglycoside- and non-aminoglycoside-adapted populations were determined relative to the average of those of three wild-type controls. The membrane potential was significantly reduced in aminoglycoside-resistant populations (Wilcoxon rank-sum test $P = 0.002$). Boxplots present the median and first and third quartiles, with whiskers showing either the maximum (minimum) value or 1.5 times the interquartile range of the data. The data are based on 10 and 22 measurements for aminoglycoside- and non-aminoglycoside-adapted populations, respectively. (**B**) Substantial differences in the accumulation of the fluorescent probe Hoechst 33342 across populations adapted to different classes of antibiotics (10 evolved populations each) relative to wild-type controls. Boxplots present the median and first and third quartiles, with whiskers showing either the maximum (minimum) value or 1.5 times the interquartile range of the data, whichever is smaller (higher). The above figures are based on the results for lineages evolved in the presence of constant sublethal antibiotic concentrations. For further results, see Supplementary Figures S4 and S5.

Supplementary Table S5). Simultaneously, these populations showed elevated intracellular levels of the fluorescent probe Hoechst 33342 (Figure 4B; Supplementary Figure S5), indicating either increased porin activity or diminished efflux pump activity (Coldham *et al*, 2010). This result contrasts with the results for populations adapted to other antibiotic classes, as these populations frequently exhibited reduced intracellular levels of Hoechst 33342 (Figure 4B; Supplementary Figure S5).

The most direct evidence for antagonistic mutational effects comes from a gene involved in K$^+$ uptake (*trkH*). Mutations in *trkH* were observed in 64% of the sequenced aminoglycoside-adapted populations. The amino-acid residue affected by one of the observed mutations (T350L) is close to the ion channel and therefore was chosen for further analysis. This mutation was inserted into wild-type *E. coli*, and the inserted mutation conferred mild resistance to aminoglycosides and, simultaneously, increased the susceptibility to other classes of antibiotics, as expected (Figure 5A). Consistent with a causal role of the PMF in this negative trade-off, this mutation resulted in a diminished membrane potential and enhanced the accumulation of Hoechst dye (Figures 5B and C). In further support of the involvement of the PMF, a related regulator of K$^+$ uptake has been shown to control both the membrane potential and the multidrug susceptibility (Castaneda-Garcia *et al*, 2011).

There are further examples supporting the scenario. The list of mutated genes entails four genes (*cyoB*, *ispA*, *nuoF*, and *nuoE*) with the following remarkable combination of properties (Supplementary Table S4). First, functional connection to electron transport can be established based on the literature data, strongly suggesting that these genes influence PMF. Second, 57% of the observed mutations in these genes generate frame-shift or in frame stop-codons, and hence most likely yield proteins with compromised or no activities. Third, null mutations in these genes reduce the aminoglycoside susceptibility but enhance the sensitivity to other antibiotics.

## Collateral sensitivity is partly linked to the AcrAB efflux system

Recent studies systematically investigated the substrate specificities of all major drug transporters through deletion and overexpression over a wide range of drugs (Nishino and Yamaguchi, 2001; Girgis *et al*, 2009; Liu *et al*, 2010; Nichols *et al*, 2011). Comparison of results of chemogenomic screens and our study revealed that as high as 75% of the antibiotics showing collateral sensitivity with aminoglycosides are also substrates of the AcrAB efflux pump system. This system is member of the resistance nodulation family, and a major multidrug resistance mechanism in *E. coli*. Overexpression of this system confers resistance to a wide range of drugs and detergents, but not to aminoglycosides (Okusu *et al*, 1996; Nishino and Yamaguchi, 2001; Alekshun and Levy, 2007). A proton electrochemical potential gradient across cell membranes is the driving force for drug efflux by this system.

On the basis of these facts, we suggest that the AcrAB efflux system has a key role in the collateral-sensitivity patterns observed. More specifically, activity of this system is assumed to be impaired in aminoglycoside-resistant lines due to the presence of mutations diminishing the membrane potential. To test this hypothesis, we examined drug resistance phenotypes conferred by the AcrAB efflux system in the presence/absence of mutations in *trkH* and *cyoB*. As shown above, mutations in these genes were frequently observed in aminoglycoside-resistant lines, and we could confirm that the corresponding strains have diminished the membrane potential (Figures 4 and 5). We took advantage of the availability of a multicopy plasmid that encodes the AcrAB transporter genes of *E. coli* with the corresponding native promoters. Following protocols of a prior study (Nishino and Yamaguchi, 2001), the plasmid was transformed into wild-type and aminoglycoside-resistant mutants. We tested the corresponding changes in susceptibilities to four representative antibiotics (all of which are known

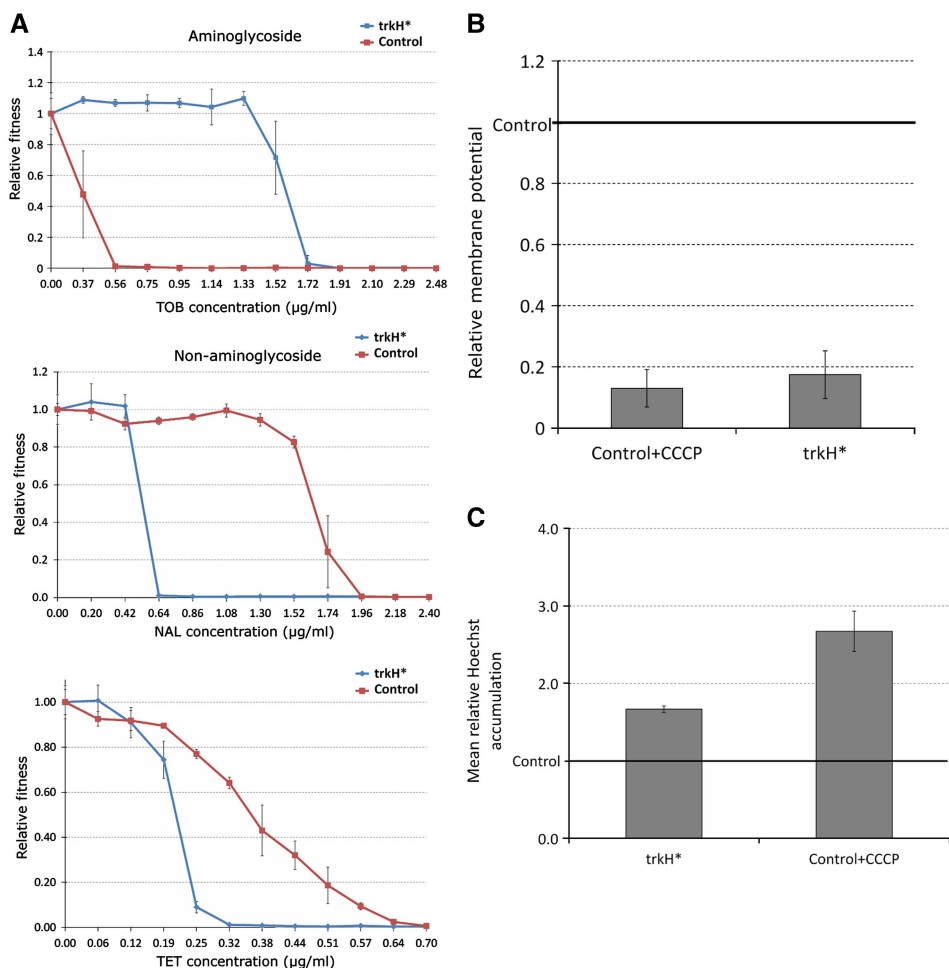

**Figure 5** Pleiotropic effects of a single mutation in trkH Individual antibiotic dose–response curves for growth inhibition were constructed for a trkH mutant strain. The red line denotes the trkH mutant strain and the blue line indicates the corresponding wild-type control. Error bars indicate the standard errors based on four technical replicates. A mutation in the *trkH* gene originally identified in a streptomycin-adapted population reduced the susceptibility to aminoglycosides but inhibited growth in the presence of several non-aminoglycoside antibiotic stresses. For more details on minimum inhibitory changes, see Supplementary Table S7. This mutation also (**B**) reduced the membrane potential (Wilcoxon rank-sum test $P = 0.02$, based on four replicate measurements) and (**C**) the enhanced accumulation of Hoechst dye (Wilcoxon rank-sum test $P = 0.0005$, based on eight replicate measurements). Control populations treated with a chemical inhibitor of the PMF (CCCP) showed similar patterns.

substrates of the AcrAB efflux pump). First, strains with deficient AcrAB efflux system were sensitive to all four antibiotics, regardless of the presence of mutations affecting the membrane electrochemical potential (Figure 6). Second, the AcrAB overexpression plasmid conferred a significant resistance to all four antibiotics in genotypes with wild-type membrane potential. Third, and most strikingly, the same plasmid conferred substantially weaker resistance when introduced into the mutant lines (Figure 6). Taken together, these results confirm that mutations conferring aminoglycoside resistance via diminishing the membrane potential increase the sensitivity to other agents by interfering with the AcrAB efflux system.

# Discussion

By combining experimental evolution, whole-genome sequencing of laboratory-evolved bacteria and biochemical assays, this work charted a map of collateral-sensitivity interactions

between antibiotics in *E. coli*, and aimed to understand these negative evolutionary trade-offs. We demonstrated that collateral-sensitivity interactions occurred at high rates. Strikingly, laboratory evolution to different aminoglycoside antibiotics frequently enhanced the sensitivity to many other antimicrobial agents (2–10 fold MIC change). Whole-genome sequencing of laboratory-evolved strains revealed multiple mechanisms underlying aminoglycoside resistance. As expected, the major targets of selection were the translational machinery, membrane transport, phospholipid synthesis, and cell envelope homeostasis. Strikingly, we also identified a broad class of mutated genes involved in maintenance of the membrane electrochemical potential. Notably, similar sets of mutations have been observed in clinical and experimental settings (Supplementary Table S4).

Aminoglycoside resistance can be achieved through reduction in the PMF across the inner membrane (Proctor *et al*, 2006; Pranting and Andersson, 2010). In this paper, we demonstrated that these changes underlie the hypersensitivity of aminoglycoside-resistant bacteria to several other antimicrobial agents,

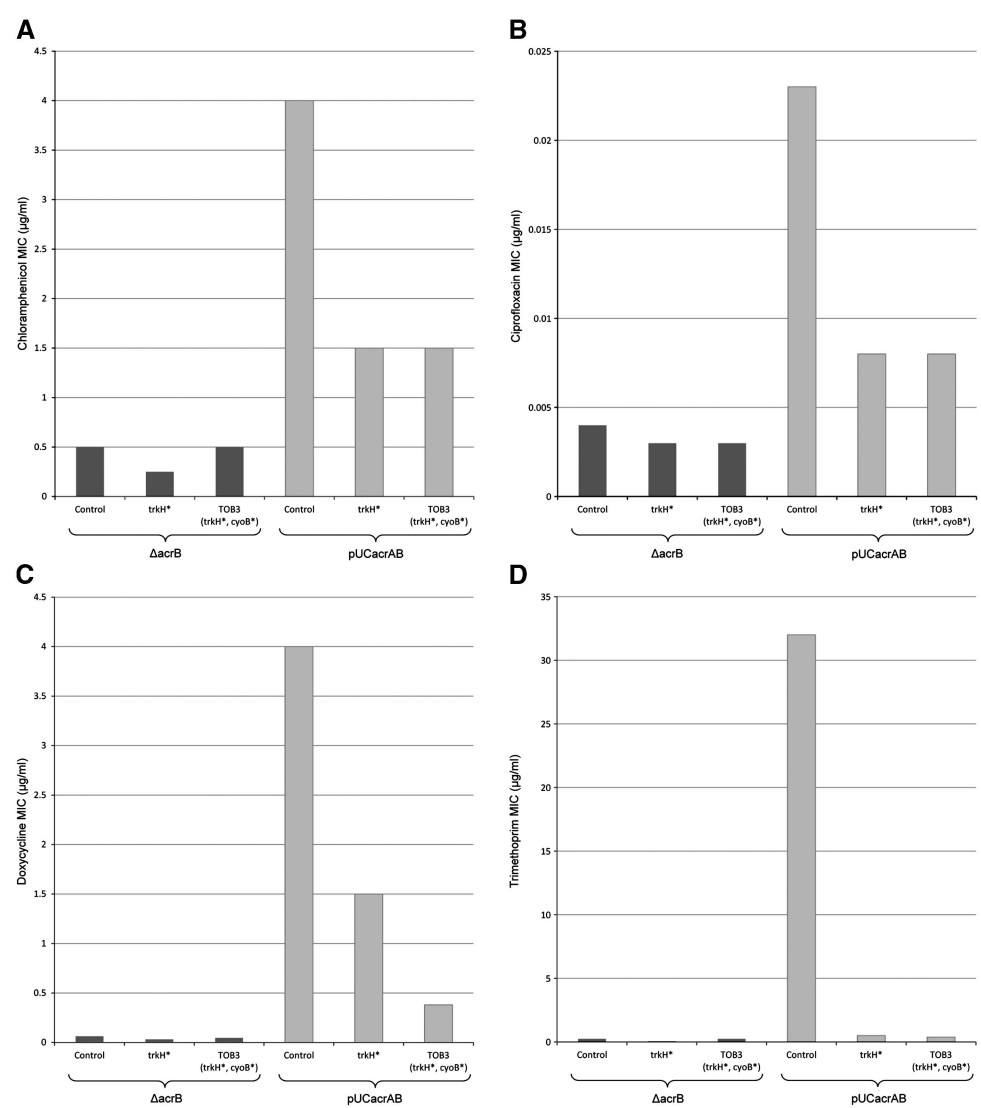

**Figure 6** Link between the copy number of the major drug efflux system AcrAB and the extent of collateral-sensitivity interactions. The AcrAB efflux system confers resistance to a variety of drugs, but not to aminoglycosides. Two aminoglycoside-resistant strains (trkH* and TOB3) and a wild-type strain (control) were modified either by deleting the *acrB* gene (ΔacrB) or by harbouring a multicopy plasmid carrying the *acrAB* genes (pUCacrAB). Change of MIC in modified strains was measured using *E*-test stripes containing one of four antibiotics (**A**) Chloramphenicol, (**B**) Ciprofloxacin, (**C**) Doxycycline, (**D**) Trimethoprim, representing different classes of modes of action (Table I). The plasmid conferred a significant resistance to all four antibiotics in control strain, but resistance levels were substantially reduced when the same plasmid was associated with membrane potential affecting mutations in either trkH (trkH*) or both *trkH* and *cyoB* genes (TOB3).

partly through diminishing the activity of PMF-dependent major efflux pumps. Taken together, these results indicate the existence of an antagonistic mechanism by which bacteria modulate intracellular antibiotic concentrations. Evolutionary experiments performed with constant and gradually increasing antibiotic concentrations yielded similar broad-scale collateral-sensitivity patterns (see Figures 1A and B), and one of the membrane potential affecting genes (*trkH*) was repeatedly mutated in response to both treatments (Supplementary Table S4). On the basis of these findings, we speculate that the membrane potential affecting mutations may arise at an early stage of resistance evolution.

Overexpression of a major multidrug transport (AcrAB) conferred only a relatively low level of resistance in association with PMF-affecting mutations. This result could have a

broad significance. PMF-dependent efflux pumps are frequently delivered by horizontal gene transfer, and have crucial contribution to the evolution of multidrug resistance patterns in a broad range of bacterial species (Paulsen *et al*, 1996; Mine *et al*, 1999; Norman *et al*, 2008). Thus, resistance to one antibiotic may not only confer changed sensitivity to another antimicrobial agent, but also affect its further evolution (see also Palmer and Kishony, 2013).

We emphasize that we do not consider our explanation exclusive. However, we failed to find evidence for other mechanisms playing a role in the observed collateral-sensitivity patterns of aminoglycoside-resistant populations. Existing chemogenomic, literature, and experimental data are fully consistent with our scenario (see Results). There is at least one potentially interesting case that should be explored in

a future work. The elongation factor fusA was regularly associated with aminoglycoside resistance in our experiments. Previous works suggest that resistance conferred by fusA mutations in Salmonella caused enhanced sensitivity to other classes of antimicrobial agents (Macvanin and Hughes, 2005). Strikingly, these mutations also caused low levels of heme biosynthesis and reduced respiratory activity.

More generally, it will be important to determine how conserved these networks of collateral-sensitivity interactions are between bacterial species. Due to the similarities in the cellular uptake mechanisms of aminoglycosides and cationic antimicrobial peptides (Moore and Hancock, 1986), our results may be more general. At least three mutated proteins (ArnC, SbmA, and TrK) in our experiments influence resistance to antimicrobial peptides (Supplementary Table S4), and more generally, mutations in the heme biosynthesis pathway in Salmonella provide resistance to several small peptides and aminoglycosides and simultaneously increase the susceptibility to other antibiotics (Pranting and Andersson, 2010).

Last, we need to emphasize the limitations of our work. First, a substantial fraction of the collateral-sensitivity interactions unrelated to aminoglycosides require mechanistic explanation (Figure 1). Second, we neglected the evolution of resistance through the acquisition of genes by horizontal transfer. Third, due to the lack of systematic studies, the frequency of PMF-altering mutations in clinical isolates is largely unknown. Therefore, discussing the direct therapeutic implications of our collateral-sensitivity network is beyond the scope of this paper. It remains controversial whether the temporal rotation or simultaneous use of two antibiotics can select against the development of resistance (Bonhoeffer et al, 1997; Chait et al, 2007). The success of such strategies may depend on the choice of antibiotics, as treatment with a single antibiotic followed by a switch to a cross-sensitive partner may represent a viable strategy. More generally, our work established the prevalence of antagonistic pleiotropy during the evolution of antibiotic resistance.

# Materials and methods

## Laboratory evolutionary experiments

We followed established protocols with minor modifications to evolve resistant bacteria by propagating them in batch cultures in the presence of antibiotics (Orlen and Hughes, 2006; Hegreness et al, 2008). Populations of Escherichia coli K12 (BW25113) were grown in autoclaved MS-minimal medium supplemented with 0.2% glucose and 0.1% casamino acids. Antibiotic solutions were prepared from powder stocks and filter sterilized before use and diluted in the growth media. Fresh antibiotic stocks were used on a weekly basis. Parallel cultures were propagated in 96-well microtiter plates, continuously shaken at ~320 r.p.m. (30°C). Plates were covered with special sandwich covers (Enzyscreen) to ensure an optimal oxygen exchange rate and limit evaporation. Every 24 h, bacterial cells were transferred by inoculating ~1–1.2 μl of stationary phase culture to 100 μl fresh medium using a 96-pin replicator (VP407) to give a daily dilution of ~100 or about 6–7 doublings. The evolved and control strains were preserved at −80°C in 20% (v/v) glycerol solution.

## Constant, sublethal antibiotic dosage

Using sublethal antibiotic concentrations (approximately half maximal inhibitory concentrations, IC50), we propagated 10 independent populations in the presence of each of the 24 antibiotics for ~140 generations (Table I), resulting in 240 parallel-evolved lines. In addition, to control for potential adaptive changes that are not specific to the employed antibiotics, we also established 10 parallel populations growing in an environment devoid of antibiotics for ~140 generations, referred to as adapted control lines. As expected, these lines showed no major changes in antibiotic susceptibility. Plates contained 36 bacteria-free wells to monitor potential contamination events: all remained uncontaminated during the entire course of the experiment. On average, the employed antibiotic had a mere 7% inhibitory effect on the growth of laboratory-adapted populations.

## Gradually increasing antibiotic dosage

In this experimental setting, populations were allowed to evolve to successively higher antibiotic concentrations. Starting with a subinhibitory (IC50) antibiotic concentration, antibiotic dosage was increased gradually (1.5 times the previous dosage) at every fourth transfer. The optical density at 600 nm (OD600) of each well was measured in a Biotek Synergy plate reader before each transfer. As expected, during the course of laboratory evolution, populations grew to different densities, reflecting independent evolutionary trajectories. Population extinction was defined as the failure to obtain growth (OD600 < 0.05). The experiments ended once only 10 populations had showed growth or antibiotic concentration had reached its upper solubility limit.

As this laboratory evolutionary protocol frequently leads to extinction of bacterial populations, 96 independent parallel populations were propagated in the presence of each antibiotic. Due to the large number of replicate lineages required, we concentrated on 12 selected antibiotics out of the 24 listed in Table I. This set still covers diverse modes of actions, but includes only 1–2 members of each major antibiotic class.

Most surviving bacterial populations from the final day of the experiments reached a very high resistance level, comparable to that found in clinical isolates (Supplementary Table S1). Depending on the antibiotics employed (and the corresponding extinction dynamics of parallel evolving populations), the experiments lasted for ~240–384 generations (Supplementary Table S1). For each antibiotic, 10 populations with the highest cell densities were selected for further analysis. We also established 10 parallel populations growing in an environment devoid of antibiotics for the same number of transfers, referred to as adapted control lines. Subsequently, we determined the antibiotic susceptibilities of these selected populations against all other antibiotics.

## Systematic measurement of antibiotic susceptibilities

Given two panels of laboratory-evolved strains, our next goal was to detect changes in their sensitivities towards other antimicrobial agents. To this end, we developed a high-throughput screening and robust statistical analysis methodology to systematically detect collateral-sensitivity interactions in E. coli.

## Growth measurement

Bacterial growth was monitored by measuring optical density (OD600) of the liquid cultures at a single time point. Preliminary experiments showed that a single reading of optical density after 14 h of incubation shows very strong linear correlation ($R^2 > 0.99$) with the area under the growth curve, a descriptor of overall inhibitory effect that covers the entire growth period (Supplementary Figure S6). We used a robotic liquid handling system (Hamilton Star Workstation) to improve reproducibility and thereby allowing us to perform hundreds of growth measurements in parallel on 384-well microtiter plates. Slight variations in temperature or humidity within the plate during incubation may lead to local trends of altered growth (within-plate effects). To overcome any measurement bias caused by the inhomogeneous environment and to convert raw OD values into relative fitness values that are comparable across plates, we employed a normalization procedure as described in Supplementary Text S1.

## Estimating collateral sensitivity

We tested the sensitivity of each evolved line against the entire set of antibiotics by measuring the growth in liquid cultures of all antibiotic-adapted lines and adapted control lines at sublethal doses of antibiotics (i.e., at around half-maximal effective concentration) in four technical replicates (i.e., strains were cultivated in quadruplicate on the same 384-well plate). In addition, we also measured the growth of evolved lines in a medium devoid of antibiotics to discern condition-specific fitness defects from general costs of resistance.

Because growth media with half-maximal effective concentrations of antibiotics are difficult to prepare in a reproducible manner, we conducted four independent experimental runs for each combination of strains and antibiotic conditions. Next, to filter out unreliable measurements and those where the antibiotic dosage was too high to detect collateral-sensitivity interactions, we excluded cases where (i) cross-contamination might have occurred on the plate during susceptibility measurements (based on the growth in non-inoculated wells), (ii) the control wells devoid of antibiotics showed large variations (coefficient of variation was above 20%), (iii) the applied antibiotic dosage was too high which strongly inhibited the growth of the adapted control populations ($>90\%$ effective concentration). This quality control procedure resulted in 2–3 replicates for each combination of strains and antibiotics.

To identify statistically significant collateral-sensitivity interactions, we compared normalized growth values of evolved lines with that of adapted control lines under the same treatment condition. Specifically, using growth data on evolved lines in each antibiotic condition, we tested whether growth of the 10 evolved lines, as a group, showed at least 10% difference from that of the 10 adapted control lines, as a group, under the same treatment condition. Statistical significance was assessed using a non-parametric bootstrap method (Efron and Tibshirani, 1994) and growth of each line was represented by the median value of the four technical replicates. The $P$-values resulting from independent experimental runs were combined using Fisher's combined probability test (data can be found in 'P_values1' sheets of Supplementary Tables S8 and S9). As a final step, we set up a rigorous statistical procedure to ensure that the collateral-sensitive interactions detected above are not due to general fitness costs of resistance (see Supplementary Text S2, data can be found in 'RF_values' and 'P_values2' sheets of Supplementary Tables S8 and S9). This yielded a matrix of evolutionary interactions between antibiotics (Supplementary Table S2). For more details on the accuracy of high-throughput interaction measurements and control for potential confounding factors, see Supplementary Text S2.

## Whole-genome resequencing

Fourteen independently evolved clones were subjected to next-generation sequencing to identify mutations responsible for the resistant phenotype. All of them showed hypersensitivity towards other classes of antibiotics, and two of them had evolved to constant, sublethal antibiotic dosage. Briefly, genomic DNA (gDNA) was extracted from selected E. coli isolates (SIGMA GenElute Bacterial Genomic DNA kit, standard procedures), fragmented, and SOLiD sequencing adaptors were ligated. Subsequently, sequencing beads were prepared and sequenced on the SOLiD System. As a result, millions of short reads (50 or 75 bp) were generated along with data indicating the sequencing quality of each nucleotide. Finally, variants, such as SNP or multi-nucleotide polymorphism (MNP), insertions and deletions (InDels), were identified compared with the reference E. coli genome using standard bioinformatics analysis.

Preparation of the libraries and sequencings were performed by cycled ligation sequencing on a SOLiD 5500xl System (Life Technologies; LT) using reagents and protocols provided by LT. Briefly, 3 μg of purified bacterial gDNA was fragmented by the Covaris S2 System to 150–350 bp. The fragmented DNA was end-repaired and ligated to P1 (5′-CCACTACGCCTCCGCTTTCCTCTCTATGGGCAGTCGGTGAT-3′) and P2 (5′CTGCCCCGGGTTCCTCATTCTCTGTGTAAGAGGCTGCTGACGGC CAAGGCG-3′) adapters, which provide the primary sequences for both amplification and sequencing of the sample library fragments. The P2 adapter contains a 10-bp barcode sequence which provided the basis

for multiplex sequencing. The templates were clonally amplified by emulsion PCR (ePCR) with P1 primer covalently attached to the bead surface. Emulsions were broken with butanol, and ePCR beads enriched for template-positive beads by hybridization with P2-coated capture beads. Template-enriched beads were extended at the 3′ end in the presence of terminal transferase and 3′ bead linker. Beads with clonally amplified DNA were then deposited onto a SOLiD Flowchip (Ondov et al, 2010). About 250 million beads with clonally amplified DNA were deposited onto one lane of the flowchip. The slide was then loaded onto a SOLiD 5500xl instrument and the 50-base sequences were obtained according to the manufacturer's protocol.

## Further analysis of genome sequences

The obtained sequences were aligned to the E. coli str. K-12 substr. MG1655chromosome (Accession NC_000913; Version NC_000913.2 GI: 49175990). Alignment was performed using Genomics Workbench (Floratos et al, 2010) 4.9 and the Omixon Gapped SOLiD Alignment 1.3.2 plugin, provided by CLC Bio and Omixon, respectively. A minimum average coverage of 50-fold was accomplished for each strain. The maximum gap and mismatch count within a single read was set to 2 with a minimum of 4 reads to call a potential variation before further analysis. Selected putative variants (SNPs and indels) detected by whole-genome resequencing were verified by PCR followed by Sanger sequencing on a 3500 Series Genetic Analyzer (LT). The primers were designed using Genomics Workbench and are available on request.

## Hoechst dye (H33342 bisbenzimide) accumulation assay

To estimate changes in cellular permeability, we implemented a recently developed and scalable fluorescence assay (Coldham et al, 2010). The method is based on accumulation of the fluorescent probe Hoechst (H) 33342 (bisbenzimide). All laboratory-evolved populations were cultured overnight in MS-minimal medium supplemented with 0.2% glucose and 0.1% casamino acid. Optical densities of evolved bacterial populations were adjusted to OD600 = 0.3. In all, 180 μl aliquots of bacterial cultures were transferred to 96-well microtiter plates (8 technical replicates per evolved line). Plates were incubated in a Synergy 2 microplate reader at 30°C, and 25 μM Hoechst dye (SIGMA) was added to each well using an auto-injection device (BioTek dispenser box). The OD and fluorescence curves were measured for 1 h with 75-s delays between readings. The first 15 data points were excluded from further analysis due to the high standard deviation between replicates. Blank normalized OD values were calibrated as described in Supplementary Text S1. Data curves were smoothed and fluorescence per OD ratio curves were calculated. Next, areas under these ratio curves were determined. Finally, we calculated changes in Hoechst dye accumulations relative to the appropriate wild-type controls derived from the same experiment.

## Measurement of bacterial membrane potential

Two adapted strains per each antibiotic selection regime were selected randomly, and were subjected to membrane potential measurement. The BacLight BacterialMembrane Potential Kit (B34950, Invitrogen) was used to assess changes in PMF in the evolved strains. Briefly, DiOC2 exhibits green fluorescence in all bacterial cells, but the fluorescence shifts towards red emission in cells maintaining high membrane potential. The ratio of red to green fluorescence provides a measure of membrane potential that is largely independent of cell size.

Overnight bacterial cultures were diluted to $\sim 10^6$ cells/ml in filtered buffer (PBS). Aliquots of 200 μl bacterial suspension were added to 96-well microtiter plates for staining treatments. The DiOC2 dye was added to each sample in a 0.03-mM concentration (no antibiotic was added to the medium). After 30 min of incubation, samples were diluted 10-fold, and were analysed by a GUAVA EasyCyte 8HT Capillary Flow Cytometer. The instrument settings were adjusted according to the BacLight kit manual. In all, 15 000 events were recorded and gated

out by visual inspection using the forward versus side scatter before data acquisition. The red/green fluorescence values for a representative set of aminoglycoside and non-aminoglycoside evolved populations were calculated relative to the average of three control wild-type populations.

## MIC and dose–response curve measurements

MICs were determined using a standard linear broth dilution technique (Wiegand *et al*, 2008). In order to maximize reproducibility and accuracy, we used a robotic liquid handling system (Hamilton Star Workstation) to prepare 12 linear dilution steps automatically in 96-well microtiter plates. Approximately $10^6$ bacteria/ml were inoculated into each well with a 96-pin replicator, and were propagated at $30°C$ shaken at 300 r.p.m. (4 replicates per strain/antibiotic concentration). After 24 h of incubation, raw OD values were measured in a Biotek Synergy 2 microplate reader. MIC was defined by a cutoff OD value (i.e., mean $+ 2$ standard deviations of OD values of bacteria-free wells containing only growth medium).

## Allele replacements

Allele replacement in trkH was constructed by a suicide plasmid-based method in a markerless allele replacement, which can be distinguished by sequencing of the given chromosomal region. For details on primer sequences, see Feher *et al* (2008). Standard steps and plasmids (pST76-A, pSTKST) of the procedure have been described (Feher *et al*, 2008). In brief, an ~800-bp long targeting DNA fragment carrying the desired point mutation in the middle was synthesized by PCR, then cloned into a thermosensitive suicide plasmid. The plasmid construct was then transformed into the cell, where it was able to integrate into the chromosome via a single crossover between the mutant allele and the corresponding chromosomal region. The desired cointegrates were selected by the antibiotic resistance carried on the plasmid at a non-permissive temperature for plasmid replication. Next, the pSTKST helper plasmid was transformed, then induced within the cells, resulting in the expression of the I-SceI meganuclease enzyme, which cleaves the chromosome at the 18-bp recognition site present on the integrated plasmid. The resulting chromosomal gap is repaired by way of RecA-mediated intramolecular recombination between the homologous segments in the vicinity of the broken ends.

## AcrAB efflux system and collateral sensitivity

To investigate the drug resistance phenotype conferred by the AcrAB efflux system, three different strains were modified either by deleting the *acrB* gene or by transforming a multicopy plasmid carrying the AcrAB efflux pump (pUCacrAB) into various strains. The three selected strains were the following: (i) the ancestral BW25113 strain (control), (ii) an aminoglycoside-evolved line carrying mutations in the *trkH* and *cyoB* genes (TOB3, see Supplementary Table S4), and (iii) the ancestral strain carrying a mutation in trkH (T350L; the corresponding strain will be referred to as trkH*). The appropriate acrB deletion strains (ΔacrB/control, ΔacrB/trkH*, ΔacrB/TOB3) were constructed using standard protocols by P1 transduction (Baba *et al*, 2006; Miller, 1972).

To increase the copy number of the efflux pump, the plasmid pUCacrAB with the corresponding native promoters was transformed into the appropriate strains (pUCacrAB/control, pUCacrAB/trkH*, pUCacrAB/TOB3) following the protocols of a prior study (Nishino and Yamaguchi, 2001). The pUCacrAB plasmid was constructed and provided us by Kunihiko Nishino and Akihito Yamaguchi (Osaka University, Osaka, Japan).

Changes in susceptibility towards four representative antimicrobial agents (chloramphenicol, ciprofloxacin, trimethoprim, and doxycycline) were tested applying E-test stripes (bioMerieux). E-test inoculum preparation and plating, strip application, and subsequent MIC determinations were carried out in accordance with the manufacturer's instructions. The applied antibiotics represent different classes of mode of action; however, the AcrAB efflux system is known to cause resistance towards all of them.

## Supplementary information

## Acknowledgements

We would like to thank Stephen G Oliver and Olivier Tenaillon for their insightful comments on an earlier version of this manuscript. We are also grateful to Dr Nishino for sending us plasmids used in this study. We also acknowledge the use of the NGS platform established by the ERC Advanced Grant 'SymBiotics' of Eva Kondorosi. This work was supported by grants from the European Research Council (202591) and the EMBO Young Investigator Programme (to CP), the Wellcome Trust, and the 'Lendület Program' of the Hungarian Academy of Sciences (to CP and BP), and the International Human Frontier Science Program Organization (to BP).

*Author contributions:* BP and CP conceived and supervised the project; VL designed the experiments, VL, RS, IN, BH, MH, BB, OM, and BC performed the experiments; GPS, VL, RB-F, GF, BS, and BK developed data analysis procedures and interpreted the data; GP gave technical support; CP and PB wrote the manuscript with contributions of all other authors.

## Conflict of interest

The authors declare that they have no conflict of interest.

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
