## [Review Process File · Molecular Systems Biology]

Bacterial evolution of antibiotic hypersensitivity

Viktória Lázár, Gajinder Pal Singh, Réka Spohn, István Nagy, Balázs Horváth, Mónika Hrtyan, Róbert Busa-Fekete, Balázs Bogos, Orsolya Méhi, Bálint Csörgő, György Pósfai, Gergely Fekete, Balázs Szappanos, Balázs Kégl, Balázs Papp and Csaba Pál

Corresponding author: Csaba Pal, Biological Research Center

Review timeline:

Submission date:	27 May 2013
Editorial Decision:	04 June 2013
Appeal letter:	06 June 2013
Editorial Decision:	04 July 2013
Revision received:	30 July 2013
Editorial Decision:	06 September 2013
Revision received:	24 September 2013
Accepted:	25 September 2013

Editor: Maria Polychronidou

Transaction Report:

1st Editorial Decision

04 June 2013

Thank you for having submitted a manuscript entitled "Evolution of antibiotic hypersensitivity" for consideration for publication in Molecular Systems Biology.

Your paper has now been seen by Editors of the Journal and we have decided to return it to you without sending it for extensive peer review.

In this study, you combine laboratory evolutionary experiments with whole-genome sequencing to investigate the collateral sensitivity interactions between antibiotics in *E. coli*. We recognize that analyzing antibiotic interactions in microorganisms is an interesting topic and we acknowledge that you propose that one of the mechanisms of antibiotic hypersensitivity related to aminoglycoside resistance, involves the loss of activity of "proton-motive force"-dependent efflux pumps. However, we feel that the conceptual novelty of these findings remains limited in view previously published studies (i.e. Sulavik et al., 2001 PMID: 11257026), demonstrating the role of efflux pumps in increased antibiotic susceptibility. Moreover, the insight into the mechanisms underlying the remaining collateral sensitivity interactions in the presented network remains limited. As such, we are not convinced that the study provides the degree of systems-level biological insight and conceptual advance our audience would expect in Molecular Systems Biology and we think that your manuscript is better suited for publication in a more specialized journal.

I am very sorry to have to disappoint you on this occasion, but I hope that this early decision will allow you to decide how to proceed with your manuscript without undue delay.

Appeal letter

06 June 2013

I wish to thank you for considering our work for publication. We felt that there could be a misunderstanding regarding the goals of our project. Most particularly, the paper cited by the editor (Sulavik et al) does not affect the originality of our work (see point 2 below). We demonstrate the existence of a link between two seemingly unrelated phenomena (A: aminoglycoside resistance and B: hypersensitivity to other drugs), and provide a simple and elegant mechanistic explanation. The novelty is not A or B, but rather the link between the two which have remained completely unnoticed so far. We are convinced that, once published, our work would be of interest to a diverse community (evolutionary biologists, systems biologists, clinicians). Let me again emphasize why we believe so:

The title of the paper says all. For the first time, we show that antibiotic hypersensitivity can evolve spontaneously under antibiotic pressure. We systematically investigate how evolutionary adaptation to antibiotic A leads to enhanced sensitivity to B. We charted the map of these evolutionary interactions across 24 antibiotics. This map has nothing to do with previous works that aimed to study antibiotic interactions based on the physiological effects of antibiotic combinations. The last work with remotely similar aims was published more than 60 years ago (Szybalski and Bryson., *Journal of Bacteriology* 1952)

For the first time, we found that such negative evolutionary trade-offs are prevalent, and 45% of them involve aminoglycosides. This is the very reason why, in this work, we focus on aminoglycosides, and do not attempt to study the remaining part of the network.

For the first time, we found a mechanistic link underlying such a negative trade-off. During laboratory evolution, aminoglycoside resistant lines accumulate mutations that lower the proton motive force, and as a by-product these mutations lower activity of major efflux pumps. The novelty is NOT that deletion of major efflux pumps enhance antibiotic susceptibility (e.g. by Sulavik et al. 2001). Needless to say, it's well described. The novelty is that reduced activity of efflux pump can spontaneously evolve as a result of genetic adaptation to a specific class antibiotic.

We demonstrate that the very same adaptive mutations that confer resistance to aminoglycosides also cause hypersensitivity to other agents. To our best knowledge, no previous evolutionary studies have suggested such a negative evolutionary trade-off during antibiotic selection.

Last, we also show that adaptation to a given antibiotic also affects evolvability towards other agents through reducing the beneficial effects of mutations. Specifically, we show that pump overexpression hardly increased resistance in aminoglycoside resistant lines. This result has potentially important clinical implications (see discussion of the paper).

Based on these points, I would be delighted if the editorial board could reconsider the decision, and submit the paper for detailed review.

2nd Editorial Decision

04 July 2013

Thank you again for submitting your work to *Molecular Systems Biology*. We have now heard back from the two referees who agreed to evaluate your manuscript. As you will see from the reports below, the referees raise concerns on your work, which should be convincingly addressed in a revision of the manuscript.

Overall, both reviewers acknowledge that the work addresses a potentially interesting topic.

However, reviewer #1 points out that the scope of this manuscript remains rather narrow, as the analysis is restricted to aminoglycoside-primed sensitivity resulting from altered proton motive force (PMF). Since the systematic mapping of collateral sensitivity/resistance and the associated mutations are presented for the first time in this study, it is essential to describe the data in more detail and provide a more comprehensive overview of the results. In particular, a deeper discussion of the map shown in Figure 1B needs to be provided, explaining the general trends underlying the observed hypersensitivity. While further experimentation for the identification of alternative mechanisms of collateral antibiotic resistance (other than changes in PMF) is not mandatory, we would strongly recommend inclusion of data that broaden the scope beyond the aminoglycosides/PMF case, if already available, as they would significantly strengthen the generality and impact of the study. Additionally, as both reviewers have requested, the plate normalization procedure needs to be described in detail.

Reviewer #1:

Project Summary

Lazar et. al report on a large scale experiment of a laboratory strain of *E. coli* adaptively evolved in a multiplexed fashion to a panel of 24 antibiotics at sub-inhibitory concentrations, and to a subset of 12 antibiotics at successively higher concentrations. Populations from both these experiments which demonstrated adaptive increases in resistance to the selected antibiotics were then cross-screened against the full panel of 24 antibiotics, generating a map of collateral cross-resistance and cross-sensitivity. The authors focus this manuscript on the analysis of the collateral hypersensitivities observed (the cross-resistance data and analysis is reserved for a separate publication), most specifically on the subset of hypersensitivities that arise collaterally with evolved aminoglycoside (AG) resistance (these represent ~45% of observed hypersensitivities). Through genome sequencing of specific mutants, biochemical characterization of the presumed primary mechanism of resistance (reduction in the proton-motive-force [PMF]), and genetic knockout / overexpression of PMF-dependent efflux pumps, the authors build a strong case for reduced PMF in AG-evolved strains giving rise to reduced PMF-driven efflux of other antibiotics, resulting in hypersensitivity. Overall, this conclusion is well-supported by the data and of general interest to the field. Although the authors collected a wealth of hypersensitivity data, this report focuses almost exclusively on AG-primed sensitivity (and really, most specifically on the subset of hypersensitive clones with altered PMF). There is potential for a much more comprehensive story (or set of stories) from these data, and this referee encourages the authors to pursue these leads (although such pursuit may, in some cases, be beyond the scope of this manuscript).

Criticisms/Clarifications

From a systems biology perspective, the topic of this manuscript is of high general interest, however in practice the scope of the experiments does not extend beyond AG-induced changes to PMF. The manuscript would be strengthened, and more adequately target Molecular Systems Biology, if experiments were extended to either a) investigate alternative mechanisms of AG-induced hypersensitivity, or b) probe mechanistic trends underlying rest of the hypersensitivity data depicted in figs. 1B and 1C. Specific lines of investigation the authors may want to consider including:

- Are there any common evolutionary trajectories (or trends) to the order of mutations observed in clones subjected to increasing drug concentrations (average 8 muts/strain)?
- Investigating the effect of observed mutants beyond *trkH*. This could entail site-specific mutations, or the creation of gene knockouts, that target the genes/complexes identified via whole-genome sequencing (i.e. those in table S5). The most interesting candidate(s) would be those genes that generate AG resistance via a PMF-orthogonal mechanism and which are observed most frequently (e.g. *fusA*). Are any of these mutants contributing to the observed hypersensitivity, or are they all passenger mutations? If they contribute, what other mechanisms of AG-induced hypersensitivity exist?
- Analyzing hypersensitivity trends, and underlying mechanisms, beyond the AGs. For instance, (based on deconvoluting Fig 1B) why is hypersensitivity to Nitrofurantoin observed so readily?

Why does evolved AG resistance commonly result in collateral hypersensitivity to the beta-lactam ampicillin, but rarely cefoxitin (another beta-lactam, albeit of the cephalosporin class)? Is there anything known about the different effect of PMF activity on the import or efflux of those drugs, despite them attacking the same/similar targets? In another vein, are there any observed/known differences between the cidal versus static antibiotics?

* An additional concern I have is not with the experimental scope, but rather the statistics behind the plate-effect normalization procedures the authors employed (text S1). There is an incredibly convoluted set of data transformations described to convert the raw OD data into the data used to calculate growth trends and relative fitness. I was confused by the methods described in text S1 (e.g. kriging) and cannot critically evaluate whether such transformations are appropriate (I fully admit that I am not an expert in these specific analyses). I recommend a reviewer who is a field expert should be given access to some or all the raw data and the Matlab code for the analytics performed, and requested to verify the necessity and accuracy of these transforms.

Recommendation

Given the dataset generated, the authors had the capacity to pursue a much more comprehensive analysis of evolved resistance, and concordant hypersensitivity. As it stands, the authors instead elected to focus on a small, but interesting aspect of the larger dataset: AG-evolved changes to *E. coli*'s PMF, and the resulting hypersensitivities to drugs known as substrates of PMF-dependent efflux pumps. Despite the narrow scope, the authors performed exemplary mechanistic follow-up with regards to the PMF-driven hypersensitivity: using appropriate biochemical (Figs 3 and 4) and genetic (Fig 5) evidence to support their WGS-guided hypothesis. It is therefore fit for publication, though would certainly be strengthened with additional experimentation.

Reviewer #2:

The manuscript entitled "Evolution of antibiotic hypersensitivity", describes an adaptive evolution study of numerous single antibiotic stress conditions, and an analysis of the interesting result where evolution to overcome aminoglycoside stress produced hypersensitivity to non-related antibiotics. The scale of the study was impressive, and the investigation of PMF-dependence of the phenotype was well conceived and executed. This reviewer especially appreciated the experiments with *trkH*, demonstrating that this mutation has the ability to reconstitute some evolved effects. Given the importance of antibiotic resistance and the thoroughness of the study, this reviewer believes that this manuscript will be suitable for publication in *Molecular Systems Biology* once the below concerns have been addressed.

1. The plate normalization procedure in Supplemental Text 1 needs further description. The raw data appear to be processed extensively, and as it stands readers are not given enough information to replicate the procedure.
2. It would be beneficial to add the drug sensitivity information to Table S4, so readers would know which mutations correspond to which antibiotic sensitivity phenotypes (perhaps columns of the MIC change).
3. In general, depletion of PMF would decrease growth-rate, and the authors discuss how they distinguished growth rate differences in the absence of antibiotics from those in the presence of antibiotics for their sensitivity measurements. However, more slowly growing strains are generally more resistant to beta lactams, but the authors do not see this. Can they offer an explanation?
4. What "fitness" was used for Supplemental Figure S1, and discussed in Text S2?

Detailed Response to Reviewers.

Reviewer #1 (Remarks to the Author):

Lazar...Pal, Evolution of antibiotic hypersensitivity, MSB,

Project Summary

Lazar et. al report on a large scale experiment of a laboratory strain of E. coli adaptively evolved in a multiplexed fashion to a panel of 24 antibiotics at sub-inhibitory concentrations, and to a subset of 12 antibiotics at successively higher concentrations. Populations from both these experiments which demonstrated adaptive increases in resistance to the selected antibiotics were then cross-screened against the full panel of 24 antibiotics, generating a map of collateral cross-resistance and cross-sensitivity. The authors focus this manuscript on the analysis of the collateral hypersensitivities observed (the cross-resistance data and analysis is reserved for a separate publication), most specifically on the subset of hypersensitivities that arise collaterally with evolved aminoglycoside (AG) resistance (these represent ~45% of observed hypersensitivities). Through genome sequencing of specific mutants, biochemical characterization of the presumed primary mechanism of resistance (reduction in the proton-motive-force [PMF]), and genetic knockout / overexpression of PMF-dependent efflux pumps, the authors build a strong case for reduced PMF in AG-evolved strains giving rise to reduced PMF-driven efflux of other antibiotics, resulting in hypersensitivity. Overall, this conclusion is well-supported by the data and of general interest to the field.

Thank you.

Although the authors collected a wealth of hypersensitivity data, this report focuses almost exclusively on AG-primed sensitivity (and really, most specifically on the subset of hypersensitive clones with altered PMF). There is potential for a much more comprehensive story (or set of stories) from these data, and this referee encourages the authors to pursue these leads (although such pursuit may, in some cases, be beyond the scope of this manuscript).

There are two separate problems here: a) The causes of hypersensitivity unrelated to aminoglycosides, and b) mechanisms underlying hypersensitivity in aminoglycoside resistant strains.

As regards a), we are happy to admit there's much left to be explored through additional experiments/analyses. However, as the reviewer rightly points out, it would go well beyond the scope of the current paper. As we don't want to go into wild speculations, we added a short general analysis on the general properties of the collateral sensitivity network only.

We also wish to highlight that we had very good reasons to concentrate on aminoglycosides: over 40% of the hypersensitivity data is related to these antibiotics. Thus, a simple pattern emerges from a complex dataset. This happens very rarely: high-throughput or genomic studies often struggle to explain a far more modest fraction of the variation observed.

As regards b), neither the reviewer nor us (after a year of serious thinking) could come up with alternative scenarios. And there's no reason why we should do so: the data suggests that altered PMF can satisfactorily explain the hypersensitivity trends observed in aminoglycoside resistant lines (see below).

Criticisms/Clarifications

From a systems biology perspective, the topic of this manuscript is of high general interest, however in practice the scope of the experiments does not extend beyond AG-induced changes to PMF.

This is a fact that we should not be ashamed of. One novel hypothesis emerged from the data and it can explain a large fraction of the evolutionary interactions.

The manuscript would be strengthened, and more adequately target Molecular Systems Biology, if experiments were extended to either a) investigate alternative mechanisms of AG-induced hypersensitivity, or b) probe mechanistic trends underlying rest of the hypersensitivity data depicted in figs. 1B and 1C. Specific lines of investigation the authors may want to consider including:

- Are there any common evolutionary trajectories (or trends) to the order of mutations observed in clones subjected to increasing drug concentrations (average 8 muts/strain)?

A comprehensive answer to this question would require sequencing of populations at intermediate stages of laboratory evolution. We avoided such experiments, not least because of previous efforts in similar directions (see e.g. Toprak et al. Nature Genetics 2012). Moreover, we have reasons to believe that hypersensitivity generating mutations may arise at an early stage of antibiotic adaptation in our experiments (i.e. under low antibiotic pressure). We briefly mention it in the discussion as follows:

“Evolutionary experiments performed with constant and gradually increasing antibiotic concentrations yielded similar broad-scale collateral sensitivity patterns (see Figure 1A and B), and one of the membrane potential affecting genes (*trkH*) was repeatedly mutated in response to both treatments (Supplementary Table S4). Based on these findings, we speculate that membrane potential affecting mutations may arise at an early stage of resistance evolution”.

- Investigating the effect of observed mutants beyond *trkH*. This could entail site-specific mutations, or the creation of gene knockouts, that target the genes/complexes identified via whole-genome sequencing (i.e. those in table S5).

Even without performing additional experiments, we already have compelling evidence that PMF-altering mutations dominate. Through analyses of chemogenomic datasets, we found four other membrane potential affecting genes, which further support our scenario. We write:

“There are further examples supporting the scenario. The list of mutated genes entails four genes (*cyoB*, *ispA*, *nuoF*, *nuoE*) with the following remarkable combination of properties (Supplementary Table S4). First, functional connection to electron transport can be established based on literature data, strongly suggesting that these genes influence PMF. Second, 57% of the observed mutations in these genes generate frame-shift or in frame stop-codons, and hence most likely yield proteins with compromised or no activities. Third, null mutations in these genes reduce aminoglycoside susceptibility but enhance sensitivity to other antibiotics”.

The most interesting candidate(s) would be those genes that generate AG resistance via a PMF-orthogonal mechanism and which are observed most frequently (e.g. *fusA*). Are any of these mutants contributing to the observed hypersensitivity, or are they all passenger mutations? If they contribute, what other mechanisms of AG-induced hypersensitivity exist?

Indeed, we found that *fusA* is frequently mutated in AG resistant lines. But whether adaptive mutations in this gene confer hypersensitivity to other agents is an unresolved issue. Even if they do, the mechanism may be very similar. We briefly discuss this problem in the discussion of the paper:

“We emphasise that we do not consider our explanation exclusive. However, we failed to find evidence for other mechanisms playing role in the observed collateral sensitivity patterns of aminoglycoside resistant populations. Existing chemogenomic, literature and experimental data are fully consistent with our scenario (see results). There is at least one potentially interesting case that should be explored in a future work. The elongation factor *fusA* was regularly associated to aminoglycoside resistance in our experiments. Previous works suggest that resistance conferred by *fusA* mutations in *Salmonella* caused enhanced sensitivity to other classes of antimicrobial agents (Macvanin & Hughes 2005). Strikingly, these mutations also caused low levels of heme biosynthesis and reduced respiratory activity”.

• Analyzing hypersensitivity trends, and underlying mechanisms, beyond the AGs.

We incorporated new analyses at the start of the result section to describe the observed trends in more detail. There’s one particularly convincing trend: antibiotic pairs with similar modes of action do not show collateral sensitivity. We write:

“The maps based on the evolutionary experiments performed with constant and gradually increasing antibiotic concentrations were similar. 85% of the interactions between antibiotics overlapped ($P < 10^{-5}$, randomization test). Three main patterns emerge from our map. First, these interactions occurred frequently: at least 35% of all investigated antibiotic pairs showed collateral sensitivity in at least one direction. Second, the mode of antibiotic action has a strong influence on the distribution of

interactions. Collateral sensitivity never occurred between antibiotic pairs that target the same cellular subsystem (Fisher exact-test, $P=0.013$). Thanks to systematic chemogenomic studies, the mode of antibiotic action can be defined and compared in a more quantitative manner. Specifically, a previous study exposed a nearly complete mutagenized *E. coli* library to several antibiotics and determined the fitness contribution of individual genes (Girgis et al, 2009). Using this dataset, we calculated the sets of genes which influence susceptibility for each antibiotic used in our study (chemogenomic profile). Collateral sensitivity was depleted between antibiotic pairs with substantial overlap in their chemogenomic profiles (Figure 1C). Third, most antibiotic classes displayed collateral sensitivity with relatively few other classes (Figure 1D)".

For instance, (based on deconvoluting Fig 1B) why is hypersensitivity to Nitrofurantoin observed so readily?

We readily admit that we don't know the answer.

Why does evolved AG resistance commonly result in collateral hypersensitivity to the beta-lactam ampicillin, but rarely cefoxitin (another beta-lactam, albeit of the cephalosporin class)? Is there anything known about the different effect of PMF activity on the import or efflux of those drugs, despite them attacking the same/similar targets?

We also noticed this trend. However, it appears to be rather weak when populations adapted to high antibiotic dosages were considered (Figure 1B). Therefore, to err on the side of caution, we do not wish to mention this pattern in the current manuscript. We do have a speculative explanation though, and it relates to impaired activity of the AcrAB pump. Apparently, the effect of AcrAB on resistance is more pronounced with antibiotics containing more lipophilic side chains (PMID: 9721312). As expected, cefoxitin has a substantially lower lipophilic coefficient than that of ampicillin. If the reviewer insists, we are happy to mention this possibility in the manuscript.

In another vein, are there any observed/known differences between the cidal versus static antibiotics?

We checked it, and the answer was a clear no. Briefly AB pairs were divided into three categories based on the killing properties of the constituting antibiotics (cidal-static, cidal-cidal, static-static), and found that after controlling for aminoglycosides, the frequency of collateral sensitivity does not differ (Chi-square $P=0.9$). We don't see how this negative result would add to the main message of the paper.

* An additional concern I have is not with the experimental scope, but rather the statistics behind the plate-effect normalization procedures the authors employed (text S1). There is an incredibly convoluted set of data transformations described to convert the raw OD data into the data used to calculate growth trends and relative fitness. I was confused by the methods described in text S1 (e.g. kriging) and cannot critically evaluate whether such transformations are appropriate (I fully admit that I am not an expert in these specific analyses). I recommend a reviewer who is a field expert should be given access to some or all the raw data and the Matlab code for the analytics performed, and requested to verify the necessity and accuracy of these transforms.

We had good reasons to employ complex data pre-processing steps. The main goal was to identify collateral sensitivity with great precision and on a large scale. Thus, the scope of the analyses demanded highly accurate measurements while minimizing systematic errors that are inevitable in high-throughput screens. Importantly, as we now show in an additional Supplementary Figure, pilot experiments demonstrated that these normalization steps successfully reduced measurement errors (Supplementary Figure 8). Furthermore, subsets of collateral sensitivity interactions obtained from the high-throughput survey were successfully validated by detailed dose-response curve analyses (Figure 2) and colony size measurements (Supplementary Figure S1), respectively. We now provide a more detailed explanation of our normalization procedure in the Supplementary Text, including a justification of the use of Gaussian process regression (kriging).

Reviewer #2 (Remarks to the Author):

The manuscript entitled "Evolution of antibiotic hypersensitivity", describes an adaptive evolution study of numerous single antibiotic stress conditions, and an analysis of the interesting result where evolution to overcome aminoglycoside stress produced

hypersensitivity to non-related antibiotics. The scale of the study was impressive, and the investigation of PMF-dependence of the phenotype was well conceived and executed. This reviewer especially appreciated the experiments with *trkH*, demonstrating that this mutation has the ability to reconstitute some evolved effects. Given the importance of antibiotic resistance and the thoroughness of the study, this reviewer believes that this manuscript will be suitable for publication in *Molecular Systems Biology* once the below concerns have been addressed.

Thank you.

1. The plate normalization procedure in Supplemental Text 1 needs further description. The raw data appear to be processed extensively, and as it stands readers are not given enough information to replicate the procedure.

Done. See response to Reviewer 1 above.

2. It would be beneficial to add the drug sensitivity information to Table S4, so readers would know which mutations correspond to which antibiotic sensitivity phenotypes (perhaps columns of the MIC change).

Done

3. In general, depletion of PMF would decrease growth-rate, and the authors discuss how they distinguished growth rate differences in the absence of antibiotics from those in the presence of antibiotics for their sensitivity measurements. However, more slowly growing strains are generally more resistant to beta lactams, but the authors do not see this. Can they offer an explanation?

We have no clear explanation, as we see exactly the opposite trend. Slow growing aminoglycoside resistant strains are hypersensitive to beta lactams. We offer a very general theory why it should be so.

4. What "fitness" was used for Supplemental Figure S1, and discussed in Text S2?

Fitness corresponds to the optical density (OD 600) of the liquid cultures at a single time point (after 14 hours). Preliminary experiments showed that a single reading of optical density after 14 hours of incubation shows very strong linear correlation ($R^2 > 0.99$) with the area under the growth curve, a descriptor of overall inhibitory effect that covers the entire growth period.

Thank you again for submitting your work to Molecular Systems Biology. We have now heard back from the referee who accepted to evaluate the study. As you will see, the referees is now supportive and we I am pleased to inform you that your study will be accepted for publication in Molecular Systems Biology, pending the following minor points:

- we would kindly ask you to provide the dataset of fitness measurements used to determine the map of collateral effects. Please include the dataset as Excel or csv file in a form that allows others to reproduce the statistical data analysis pipeline used in this study.

Reviewer #1:

In my original review of this manuscript, my primary concerns were A) the authors had a narrow-scope focus for AG-primed PMF hypersensitivity and B) their stats were unintelligible to the lay scientist.

Regarding A): I gave them an out by essentially suggesting "It'd be nice if you did more analyses (e.g. x,y,z), but that may be out of scope". In the revision, the authors did do some nice, simple global hypersensitivity analyses that address the flavor of my criticism. Where asked for time-consuming follow-up experiments, the authors willingly opted for the "out", and understandably so. Regarding B): The new supplement does a much, much, much better job of making their statistical transforms understandable.

With these points in mind, I support publication of this revision in MSB.

We were delighted to see the final decision that our study will be accepted for publication. Regarding to the minor points, we included a new Supplementary Table S8 and S9 containing the mean relative fitness values (as explained in the Materials and Methods) calculated from our high-throughput fitness measurements. Please let us know if you need anything else to proceed.